# CircNet: Meshing 3D Point Clouds with Circumcenter Detection

**Huan Lei, Ruitao Leng, Liang Zheng, Hongdong Li**
School of Computing, The Australian National University

## Abstract

Reconstructing 3D point clouds into triangle meshes is a key problem in computational geometry and surface reconstruction. Point cloud triangulation solves this problem by providing edge information to the input points. Since no vertex interpolation is involved, it is beneficial to preserve sharp details on the surface. Taking advantage of learning-based techniques in triangulation, existing methods enumerate the complete combinations of candidate triangles, which is both complex and inefficient. In this paper, we leverage the *duality* between a triangle and its circumcenter, and introduce a deep neural network that detects the circumcenters to achieve point cloud triangulation. Specifically, we introduce multiple anchor priors to divide the neighborhood space of each point. The neural network then learns to predict the presences and locations of circumcenters under the guidance of those anchors. We extract the triangles dual to the detected circumcenters to form a primitive mesh, from which an edge-manifold mesh is produced via simple post-processing. Unlike existing learning-based triangulation methods, the proposed method bypasses an exhaustive enumeration of triangle combinations and local surface parameterization. We validate the efficiency, generalization, and robustness of our method on prominent datasets of both watertight and open surfaces. The code and trained models are provided at https://github.com/Ruitao-L/CircNet.

## 1 Introduction

Point cloud triangulation (Cazals & Giesen 2004) aims at reconstructing triangle meshes of object surfaces by adding edge information to their point cloud representations. The input point clouds are usually produced by either scanning sensors (*e.g.*, LiDAR) or surface sampling methods. Compared to implicit surface reconstruction (*e.g.*, Kazhdan et al. 2006), explicit triangulation has the advantages of preserving the original input points and fine-grained details of the surface. Moreover, it does not require oriented normals which are difficult to obtain in practice. Recent advances in geometric deep learning have seen widespread applications of neural functions for surface representations (*e.g.*, Park et al. 2019; Sitzmann et al. 2020b;a; Erler et al. 2020; Gropp et al. 2020; Atzmon & Lipman 2020a;b; Ben-Shabat et al. 2022; Ma et al. 2021; 2022). In comparison, only a few methods have been proposed to directly learn triangulation of point clouds by using neural networks. This is probably attributed to the combinatorial nature of the triangulation task, hindering the uptake of learning-based methods. The existing works have to enumerate combinations of candidate triangles around each input point, and use neural networks to predict their existence in the triangle mesh (Sharp & Ovsjanikov 2020; Liu et al. 2020). Figure 1(a) illustrates the local complexity of those combinatorial methods using a point with four neighboring points. Typically, for a point with $K$ neighbors, the combinatorial methods propose $\binom{K}{2}$ or $\mathcal{O}(K^2)$ candidate triangles.

Different from these methods, we propose to exploit the duality relationship between a triangle and its circumcenter to implement point cloud triangulation. That is, each vertex of a triangle is equally distant to its circumcenter. We use this characteristic to find triangle triplets from their circumcenters. Figure. 1(b) shows the duality based on the same example of Fig. 1(a). Our method recovers the vertex triplets of triangle $(\mathbf{p}, \mathbf{q}_1, \mathbf{q}_3)$ based on its circumcenter $\mathbf{c}$ and the equidistant characteristic, *i.e.* $\|\mathbf{p} - \mathbf{c}\| = \|\mathbf{q}_1 - \mathbf{c}\| = \|\mathbf{q}_3 - \mathbf{c}\|$. To obtain circumcenters for point cloud triangulation, we introduce a neural network that is able to detect the circumcenters of all triangles in a mesh. To the best of our knowledge, this is the first single-shot detection architecture for point cloud triangulation. We are inspired by the one-stage methods in object detection (*e.g.*, Liu et al. 2016).

Unlike previous combinatorial methods, the proposed method removes the requirement for candidate triangles. Specifically, we detect circumcenters in the neighborhood space of each point, under the guidance of a group of anchor priors. The neural network predicts whether a circumcenter exists in the reconstructed mesh, and where it is. We extract triangles induced by the detected circumcenters to obtain a primitive mesh. The final surface mesh is produced by enforcing edge-manifoldness and filling small holes on the primitive mesh.

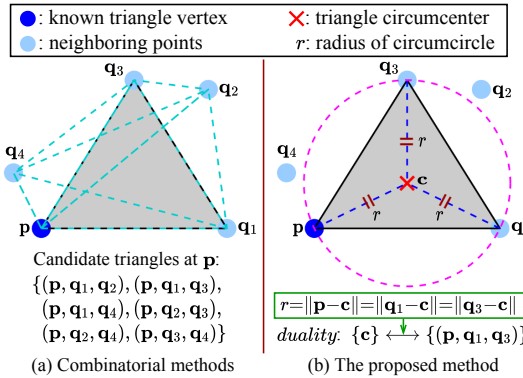

Candidate triangles at $\mathbf{p}$:
$\{(\mathbf{p}, \mathbf{q}_1, \mathbf{q}_2), (\mathbf{p}, \mathbf{q}_1, \mathbf{q}_3),$
$(\mathbf{p}, \mathbf{q}_1, \mathbf{q}_4), (\mathbf{p}, \mathbf{q}_2, \mathbf{q}_3),$
$(\mathbf{p}, \mathbf{q}_2, \mathbf{q}_4), (\mathbf{p}, \mathbf{q}_3, \mathbf{q}_4)\}$

(a) Combinatorial methods

$r = \|\mathbf{p} - \mathbf{c}\| = \|\mathbf{q}_1 - \mathbf{c}\| = \|\mathbf{q}_3 - \mathbf{c}\|$
*duality*: $\{\mathbf{c}\} \longleftrightarrow \{(\mathbf{p}, \mathbf{q}_1, \mathbf{q}_3)\}$

(b) The proposed method

Figure 1: An example of a point $\mathbf{p}$ with four neighboring points $\mathbf{q}_1, \mathbf{q}_2, \mathbf{q}_3, \mathbf{q}_4$. (a) The combinatorial methods propose all of the six triangles incident to $\mathbf{p}$ as candidate triangles. They are $(\mathbf{p}, \mathbf{q}_1, \mathbf{q}_2), (\mathbf{p}, \mathbf{q}_1, \mathbf{q}_3), (\mathbf{p}, \mathbf{q}_1, \mathbf{q}_4), (\mathbf{p}, \mathbf{q}_2, \mathbf{q}_3), (\mathbf{p}, \mathbf{q}_2, \mathbf{q}_4), (\mathbf{p}, \mathbf{q}_3, \mathbf{q}_4)$. The neural network has to classify the targeted triangle $(\mathbf{p}, \mathbf{q}_1, \mathbf{q}_3)$ out of the six candidates. (b) The proposed method eliminates the candidate proposals by detecting a circumcenter $\mathbf{c}$ and exploiting its duality with the triangle $(\mathbf{p}, \mathbf{q}_1, \mathbf{q}_3)$ to identify the targeted triangle.

To validate the proposed method, we train the detection neural network on the ABC dataset (Koch et al. 2019). The trained model is evaluated on the ABC and other datasets, including FAUST (Bogo et al. 2014), MGN (Bhatnagar et al. 2019), and Matterport3D (Chang et al. 2017). The method not only reconstructs meshes in high quality, but also outperforms the previous learning-based approaches largely in efficiency. It generalizes well to unseen, noisy and non-uniform point clouds. Our main contributions are summarized below:

- We introduce the first neural architecture that triangulates point clouds by detecting circumcenters. The duality between a triangle and its circumcenter is exploited afterwards to extract the vertex triplets of each triangle in the mesh.

- The proposed neural network is able to reconstruct primitive meshes in milliseconds, due to its single-shot detection pipeline and its removal of candidate proposals. Normals are not required.

- The proposed method casts no restriction on the surface topology, *e.g.*, being watertight. Instead, it allows the surfaces to be *open* and have genus ('hole') larger than one.

- The method generalizes well to unseen point clouds including those of large-scale scenes. It is robust to non-uniform and noisy data. These indicate its promise for real-world applications.

## 2   RELATED WORK

Point cloud triangulation and implicit surface functions are two important research directions surface reconstruction. Their major difference is that the former preserves input points, while the latter does not. Alpha shapes (Edelsbrunner & Mücke 1994) and the ball pivoting algorithm (Bernardini et al. 1999) are representatives among the traditional methods in point cloud triangulation. The Poisson surface reconstruction (Kazhdan et al. 2006; Kazhdan & Hoppe 2013) is a classical approach in implicit surface functions, but it depends on oriented normals of the input points for good performance. Marching Cubes (Lorensen & Cline 1987) and Dual Contouring (Ju et al. 2002) are restricted to extract the triangle meshes of isosurfaces from their signed distance fields. We refer interested readers to surveys Berger et al. 2017; Cheng et al. 2013; Newman & Yi 2006 for more in depth discussions of the traditional surface reconstruction methods.

### 2.1   IMPLICIT NEURAL FUNCTIONS

Implicit neural functions are techniques in geometric deep learning. They exploit the approximation power of neural networks to represent shapes and scenes as level-sets of continuous functions (Atzmon et al. 2019). Existing literature trains networks to learn either occupancy functions (Mescheder et al. 2019; Peng et al. 2020) or distance functions (Park et al. 2019; Sitzmann et al. 2020b; Atzmon & Lipman 2020a). Since the introduction of DeepSDF (Park et al. 2019), major advances in this research field include simplifying requirements on the ground-truth (*e.g.*, signs of distances) (Atzmon & Lipman 2020a;b), exploring the high-frequency features (Sitzmann et al. 2020b; Tancik et al. 2020), and improving loss functions for better surface representation (Gropp et al. 2020; Ben-Shabat

et al. 2022). Implicit functions generate volumetric occupancy or distance fields of isosurfaces, and are followed by Marching Cubes for mesh extraction. Methods in this research direction usually require oriented normals of points, and tend to oversmooth sharp details on the surface. In addition, their neural networks have to be trained with careful initialisation, and are slow at inference stage due to the dense point queries.

There are also methods extending the traditional Marching Cubes, Poisson surface reconstruction, and Dual Contouring using neural networks (Liao et al. 2018; Chen & Zhang 2021; Peng et al. 2021; Chen et al. 2022). Altogether, they demand interpolations of triangle vertices to reconstruct the mesh, and are unlikely to preserve the fine-grained details carried by the input points.

## 2.2  LEARNING-BASED POINT CLOUD TRIANGULATION

Compared to their popularity in implicit surface functions, neural networks are much less studied in point cloud triangulation. Existing methods typically reconstruct the triangle mesh by pre-establishing the complete combinations of candidate triangles or parameterizing the local surface patches into the 2D space. Sharp & Ovsjanikov (2020) propose a two-component architecture, PointTriNet. It utilizes a proposal network to recommend candidate triangles, and determines their existence in the triangle mesh with a classification network. By comparing the geodesic and Euclidean distances, Liu et al. (2020) introduce a metric IER to indicate the triangle presence in the reconstructed mesh. They train a neural network to predict the IER values of $\mathcal{O}(K^2N)$ candidate triangles. Those candidates are established offline via $K$-nearest neighbor ($K$NN) search (Preparata & Shamos 2012). Here $N$ represents the total number of points in the point cloud. As a classical method in computation geometry, Delaunay triangulation (Mark et al. 2008) guarantees point clouds to be triangulated into manifold meshes in the 2D scenario. To exploit this method, Rakotosaona et al. (2021b) extract $K$NN patches from the 3D point cloud, and parameterize them into 2D space using neural networks. However, dividing point cloud triangulation into learning-based parameterization and computational triangulation makes their method computationally expensive. Later, Rakotosaona et al. (2021a) also study to differentiate the Delaunay triangulation by introducing weighting strategies. Yet, this method is limited to triangular remeshing of manifold surfaces, and not applicable to triangulation of 3D point clouds.

Due to the local computational nature, learning-based triangulation methods generalize well to unseen point clouds of arbitrary shapes and scenes. On the other hand, the implicit neural functions are restricted to shape/surface representations in an instance or category level. It is non-trivial to apply them to cross-category data, *e.g.* mugs to elephants, cars to indoor rooms. We note that even the local implicit functions often generalize poorly (Tretschk et al. 2020; Chabra et al. 2020).

## 2.3  LEARNING TRIANGULATION VIA CIRCUMCENTER DETECTION

Departing significantly from the learning-based triangulation approaches described above, we exploit the duality between a triangle and its circumcenter, and reformulate the combinatorial triangulation as a detection problem of triangle circumcenters. This would facilitate the geomtric deep learning techniques to be applied. Superior to PointTriNet (Sharp & Ovsjanikov 2020) which necessitates a two-stage design, our method enables the usage of a one-stage detection pipeline which largely contributes to its efficiency. It has a time complexity of only $\mathcal{O}(tN)$ where $t$ indicates the number of anchors. This is significantly less than the $\mathcal{O}(K^2N)$ of IER (Liu et al. 2020). Similar to existing learning-based methods, we triangulate point clouds based on local $K$NN patches.

## 3  METHOD

Given a point cloud representation of a surface $\mathcal{P} = \{\mathbf{p}_n \in \mathbb{R}^3\}_{n=1}^N$, we focus on triangulating the point cloud into a mesh that reconstructs the underlying surface. Unlike implicit surface functions which generate new points for mesh reconstruction (Kazhdan et al., 2006), point cloud triangulation *preserves* the input points by only adding edge information to the existing points. Let $\mathcal{T} = \{(\mathbf{p}_{n_1}, \mathbf{p}_{n_2}, \mathbf{p}_{n_3}) | n_1, n_2, n_3 \in \{1, 2, \ldots, N\}, n_1 \neq n_2 \neq n_3\}$ be an optimal triangulation of $\mathcal{P}$. Typically, it reconstructs the surface as an edge-manifold mesh. This indicates that each edge in the triangulation such as $(\mathbf{p}_{n_1}, \mathbf{p}_{n_2})$ is adjacent to two triangle faces at most. In particular, edges adjacent to one face are the boundary edges.

**Overview.** Based on the local geometrics in §3.1, we detect circumcenters to predict the 1*st-order adjacent triangles* of each point. In §3.2, we introduce the default anchor priors which help to

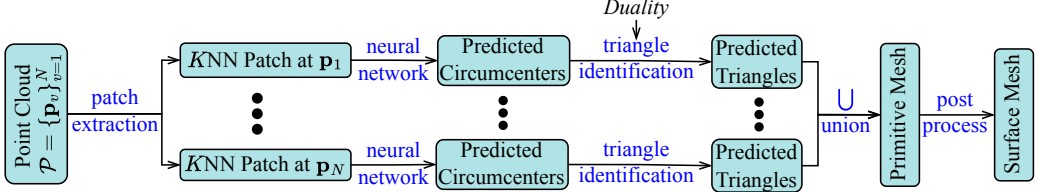

Figure 2: The triangulation process of our method for point cloud $\mathcal{P}$. We extract $K$NN patches to obtain the local geometrics of each point $\mathbf{p}_v$. The neural network detects circumcenters and identifies the adjacent triangles of each $\mathbf{p}_v$ based on their patch inputs. The union of all identified triangles forms the primitive mesh, which is post-processed into an edge-manifold surface mesh.

guide the detection of circumcenters. Details of the detection architecture are presented in §3.3. We train the neural network with multi-task loss function discussed in §3.4. During inference (§3.5), it triangulates the input point cloud efficiently into a primitive mesh. We post-process the primitive mesh to be an edge-manifold surface mesh. Figure 2 summarizes our triangulation process.

## 3.1 LOCAL GEOMETRICS

$K$**NN patch.** The local geometrics of a point contain rich information for predicting its adjacent triangles. We exploit them as the inputs to our detection network. Specifically, we extract the local geometrics of each point based on their neighborhoods. Let $\mathcal{K}(\mathbf{p}) = \{\mathbf{q}_k | \mathbf{q}_k \neq \mathbf{p}\}_{k=1}^K$ be a $K$NN patch composed of the $K$-nearest neighbor[1] ($K$NN) points of $\mathbf{p} \in \mathcal{P}$, and $d_0(\mathbf{p}) > 0$ be the distance from $\mathbf{p}$ to its nearest neighbor in $\mathcal{K}(\mathbf{p})$. To make the neural network robust to density variations in the data, we normalize the $K$NN patch $\mathcal{K}(\mathbf{p})$ using a scalar $\eta(\mathbf{p}) = \frac{\eta_0}{d_0(\mathbf{p})}$. Here $\eta_0$ is a hyperparameter controlling the spatial resolution of each $K$NN patch. The normalized patch is represented as $\overline{\mathcal{K}}(\mathbf{p}) = \{\overline{\mathbf{q}}_k | \overline{\mathbf{q}}_k = \eta(\mathbf{p}) \cdot (\mathbf{q}_k - \mathbf{p})\}_{i=1}^K$. We design graph convolution in §3.3 to learn global representations of each patch from the input geometrics $\overline{\mathcal{K}}(\mathbf{p})$ for circumcenter detection.

**Duality.** Taking $\mathcal{T}$ as the optimal triangulation of $\mathcal{P}$, we denote the adjacent triangles of point $\mathbf{p}$ as $\mathcal{T}(\mathbf{p}) = \{\mathbf{T}_i(\mathbf{p}) | \mathbf{p} \in \mathbf{T}_i(\mathbf{p})\}$, and the circumcenters of those triangles as $\mathcal{C}(\mathbf{p}) = \{\mathbf{X}_i(\mathbf{p})\}$. Our network learns to detect the circumcenters and then extract the adjacent triangles. Let $\widehat{\mathcal{C}}(\mathbf{p}) = \{\widehat{\mathbf{X}}_m(\mathbf{p})\}_{m=1}^M$, $\widehat{\mathcal{T}}(\mathbf{p}) = \{\widehat{\mathbf{T}}_m(\mathbf{p}) | \mathbf{p} \in \widehat{\mathbf{T}}_m(\mathbf{p})\}_{m=1}^M$ be their respective predictions. To extract the triangle triplets $\widehat{\mathbf{T}}_m(\mathbf{p})$ based on $\widehat{\mathbf{X}}_m(\mathbf{p})$, we follow the characteristic that the three vertices of a triangle are equidistant to its circumcenter. In practice, the equidistant characteristic has to be applied with approximations due to the imperfections of the predicted $\widehat{\mathbf{X}}_m(\mathbf{p})$. We compute the distance from $\mathbf{p}$ and each of its neighbor point $\mathbf{q}_k \in \mathcal{K}(\mathbf{p})$ to a detected circumcenter $\widehat{\mathbf{X}}_m(\mathbf{p})$ as

$$d_m(\mathbf{p}) = \|\mathbf{p} - \widehat{\mathbf{X}}_m(\mathbf{p})\|_2, \; d_m(\mathbf{q}_k) = \|\mathbf{q}_k - \widehat{\mathbf{X}}_m(\mathbf{p})\|_2. \tag{1}$$

The triangle vertices are determined by the difference between distances $d_m(\mathbf{q}_k)$ and $d_m(\mathbf{p})$. Let

$$\delta_m(\mathbf{q}_k, \mathbf{p}) = \big|d_m(\mathbf{q}_k) - d_m(\mathbf{p})\big|, \tag{2}$$

$$\Delta_m(\mathbf{p}) = \big\{\delta_m(\mathbf{q}_k, \mathbf{p}) | \mathbf{q}_k \in \mathcal{K}(\mathbf{p})\big\}. \tag{3}$$

We recover the triangle triplets by selecting the two points $\mathbf{q}_u, \mathbf{q}_v$ which induce the two smallest $\delta_m(\cdot, \mathbf{p})$ in $\Delta_m(\mathbf{p})$. Finally, the triangle associated to $\widehat{\mathbf{X}}_m(\mathbf{p})$ is identified as $\widehat{\mathbf{T}}_m(\mathbf{p}) = (\mathbf{p}, \mathbf{q}_u, \mathbf{q}_v)$.

## 3.2 ANCHOR PRIORS

We use multiple *anchor points* to partition the neighborhood space of each point into different cells. The predefined anchor points and cells guide the neural network in its detection of circumcenters. We specify the anchor points in the spherical coordinate system $(\rho, \theta, \phi)$ as it has fixed ranges along the azimuth ($\theta$) and inclination ($\phi$) directions, *i.e.* $\theta \in (-\pi, \pi]$, $\phi \in [-\frac{\pi}{2}, \frac{\pi}{2}]$. Regarding the radius ($\rho$) direction, we determine its range according to the distributions of circumcenters in the training data, denoted as $(0, R]$. With known ranges, we split the $\rho, \theta, \phi$ each uniformly using fixed steps $\Delta\rho, \Delta\theta, \Delta\phi$, respectively. This results in the number of splits for $\rho, \theta, \phi$ to be $t_\rho = \lceil \frac{R}{\Delta\rho} \rceil$,

---

[1] We assume that $K$ is large enough to cover the 1-ring neighborhood of $\mathbf{p}$ in the triangle mesh.

$t_\theta = \lceil \frac{2\pi}{\Delta\theta} \rceil$, $t_\phi = \lceil \frac{\pi}{\Delta\phi} \rceil$, and a total number of $t = t_\rho \times t_\theta \times t_\phi$ anchor points. We represent them as $\mathcal{A} = \{\mathbf{a}_j = (a_{j_1}^\rho, a_{j_2}^\theta, a_{j_3}^\phi)\}_{j=1}^t$, where $a_{j_1}^\rho, a_{j_2}^\theta, a_{j_3}^\phi$ each are explicitly defined as

$$\begin{cases} a_{j_1}^\rho = \frac{\Delta\rho}{2} + (j_1 - 1)\Delta\rho, \ j_1 \in \{1, \cdots, t_\rho\}, \\ a_{j_2}^\theta = \frac{\Delta\theta}{2} + (j_2 - 1)\Delta\theta, \ j_2 \in \{1, \cdots, t_\theta\}, \\ a_{j_3}^\phi = \frac{\Delta\phi}{2} + (j_3 - 1)\Delta\phi, \ j_3 \in \{1, \cdots, t_\phi\}. \end{cases} \quad (4)$$

For each anchor point $\mathbf{a}_j = (a_{j_1}^\rho, a_{j_2}^\theta, a_{j_3}^\phi)$, we associate it with an anchor cell defined by the partitioned space $\mathbb{I}_j^a = I_{j_1}^\rho \times I_{j_2}^\theta \times I_{j_3}^\phi$. The anchor cells play an important role in the matching of circumcenters and anchor points (§3.4). We specify the intervals $I_{j_1}^\rho, I_{j_2}^\theta, I_{j_3}^\phi$ as

$$\begin{cases} I_{j_1}^\rho = \left[a_{j_1}^\rho - \frac{\Delta\rho}{2}, a_{j_1}^\rho + \frac{\Delta\rho}{2}\right], \\ I_{j_2}^\theta = \left[a_{j_2}^\theta - \frac{\Delta\theta}{2}, a_{j_2}^\theta + \frac{\Delta\theta}{2}\right], \\ I_{j_3}^\phi = \left[a_{j_3}^\phi - \frac{\Delta\phi}{2}, a_{j_3}^\phi + \frac{\Delta\phi}{2}\right]. \end{cases} \quad (5)$$

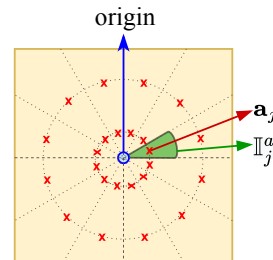

Figure 3: An example of the anchor priors in 2D. $\Delta\theta = \frac{\pi}{6}$ is used. The anchor points are plotted in red cross, and the anchor cell is colorized as green.

See Fig. 3 for an example of the anchor points and cells in 2D (*i.e.* no elevation direction). With the usage of anchor points and cells, we reduce the combinatorial triangulation of complexity $\mathcal{O}(K^2 N)$ to a dual problem of complexity $\mathcal{O}(tN)$. Empirically, $t \ll K^2$. Alternative methods for defining the anchor points can be in the Cartesian coordinate system or using the data clustering techniques (Bishop & Nasrabadi 2006). Yet, they either lead to larger $t$ and hence higher complexity, or make the anchor definition and network training complicated.

### 3.3 NETWORK DESIGN

Based on the normalized $KNN$ patch $\overline{\mathcal{K}}(\mathbf{p})$ of a point $\mathbf{p}$, we design a neural network that is able to predict the circumcenters for adjacent triangle identification. The input to our network is a *star graph* which has point $\mathbf{p}$ as its internal node and the neighborhoods $\{\mathbf{q}_k\}$ as its leaf nodes. We present the graph convolution below to encode local geometrics of $\mathbf{p}$ into a global feature representation. The depthwise separable convolution (Chollet 2017) is explored here. Let $\beta$ be the depth multiplier, $\mathbf{h}^{l-1}(\mathbf{q}_k)$ be the input features of point $\mathbf{q}_k$ at layer $l$, $C_{in}$ be the dimension of $\mathbf{h}^{l-1}(\mathbf{q}_k)$, and $C_{out}$ be the expected dimension of output features. We compute the output features $\mathbf{h}^l(\mathbf{p})$ of $\mathbf{p}$ as

$$\mathbf{h}^l(\mathbf{p}) = \sum_{k=1}^K \sum_{i=1}^\beta \mathbf{W}_{i2}\Big(\big(\mathbf{W}_{i1}\mathbf{h}^{l-1}(\mathbf{q}_k)\big) \odot \mathbf{h}^{l-1}(\mathbf{q}_k)\Big) + \mathbf{b}. \quad (6)$$

Here $\mathbf{W}_{i1}, \mathbf{W}_{i2}, \mathbf{b}$ are learnable parameters in the graph convolution. The sizes of $\mathbf{W}_{i1}, \mathbf{W}_{i2}$ are $C_{in} \times C_{in}, C_{out} \times C_{in}$ respectively, and the length of $\mathbf{b}$ is $C_{out}$. We use the graph convolution only once to calculate global feature representations of each patch. The positional encoding in (Mildenhall et al., 2020) is also employed to transform the $(x, y, z)$ coordinates into high-frequency input signals. Figure 4(b) shows the configurations of our neural detection architecture. The ultimate output of our network is a tensor of size $t \times s \times 4$, where $s$ indicates the number of circumcenters we predict in each anchor cell, and 4 contains confidence ($z$) about the existence of circumcenter in a cell and its predicted coordinates in 3D. Because the usage of one-stage pipeline, the proposed network can detect circumcenters and adjacent triangles efficiently.

### 3.4 TRAINING

We train the neural network based on multi-task loss functions, similar to those exploited in object detection (Liu et al. 2016). The binary cross-entropy is used for classifying the anchor cells, and the smooth L1 loss (Girshick 2015) is applied for localizing the circumcenters.

**Matching strategy.** To train the network accordingly, we have to match the ground-truth circumcenters to the anchor points. Thus, we transform the ground-truth circumcenters $\mathcal{C}(\mathbf{p}) = \{\mathbf{X}_i(\mathbf{p})\}$ into a spherical coordinate system centered at $\mathbf{p}$, denoted as $\mathcal{S}(\mathbf{p}) = \{(\rho_i, \theta_i, \phi_i)\}$ where $(\rho_i, \theta_i, \phi_i)$ are the spherical coordinates of $\mathbf{X}_i(\mathbf{p}) - \mathbf{p}$. For each $\mathbf{X}_i(\mathbf{p})$, we match it to the anchor point $\mathbf{a}_* = (a_{*_1}^\rho, a_{*_2}^\theta, a_{*_3}^\phi)$ if $(\rho_i, \theta_i, \phi_i) \in \mathbb{I}_*^a$. In this way, it is possible that an anchor point is matched

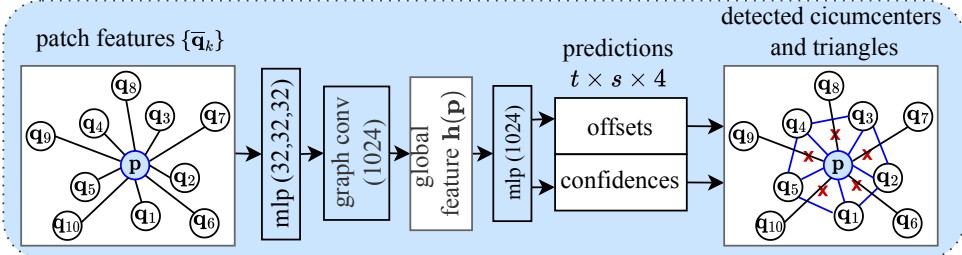

Figure 4: The neural network for circumcenter detection. It learns a global feature representation for each patch using multi-layer perceptrons and the proposed graph convolution. The global feature $\mathbf{h}(\mathbf{p})$ is used to make predictions of the circumcenters. We show in the right a toy triangulation of the input patch at $\mathbf{p}$. The detected circumcenters are in red cross and the triangle edges are in blue.

to multiple circumcenters. That is why we allow multiple circumcenters ($s$) to be detected in a single anchor cell. The proposed network detects each ground-truth circumcenter $\mathbf{X}_i(\mathbf{p}) \in \mathcal{C}(\mathbf{p})$ by predicting its parameterized offsets $\mathbf{g}_i(\mathbf{p}) = (g_i^\rho, g_i^\theta, g_i^\phi)$, defined as

$$g_i^\rho = \frac{\rho_i - a_*^\rho}{\Delta\rho}, \; g_i^\theta = \frac{\theta_i - a_*^\theta}{\Delta\theta}, \; g_i^\phi = \frac{\phi_i - a_*^\phi}{\Delta\phi}. \tag{7}$$

The predicted offsets are $(\widehat{g}_m^\rho, \widehat{g}_m^\theta, \widehat{g}_m^\phi)$. We recover its associated circumcenter prediction $\widehat{\mathbf{X}}_m(\mathbf{p}) \in \widehat{\mathcal{C}}(\mathbf{p})$ with proper coordinate transformation. Furthermore, by analyzing the distributions of circumcenters in the training set, we find that two predictions per anchor cell (*i.e.* $s = 2$) reaches a good balance between performance and efficiency.

**Binary cross-entropy loss.** For each anchor cell, whether it contains any ground-truth circumcenters or not is a binary classification task. We predict a confidence $z$ for each of them to indicate the existence of circumcenters. It can be seen from §3.2 that we define the anchor points *compactly* in the neighborhood space of a point. In reality, the circumcenters of adjacent triangles for each individual point distribute closely around the surface manifold. Such facts result in the majority of anchor cells to be unoccupied. They comprise the negative samples $\mathcal{N}_{neg}$ in our classification, while the occupied cells comprise the positive samples $\mathcal{N}_{pos}$. Assume $|\mathcal{N}_{pos}| = N_p$, $|\mathcal{N}_{neg}| = N_n$, and $p$ is the probability correlated to the confidence $z$, we compute the binary cross-entropy as

$$\mathcal{L}_1 = -\frac{1}{N_p}\sum_{i=1}^{N_p}\log(p_i) - \frac{1}{N_n}\sum_{i=1}^{N_n}\log(1-p_i), \;\; \text{where } p_i = \frac{1}{1+\exp(-z_i)}. \tag{8}$$

We employ hard negative mining for better training (Liu et al. 2016).

**Smooth L1 loss.** As mentioned above, we predict $s = 2$ circumcenters inside each cell. Let $\mathcal{G}_i = \{\mathbf{g}_{i1}, \cdots, \mathbf{g}_{i\tau}\}$ be the ground-truth offset coordinates of all circumcenters in the $i$th cell, and $\{\widehat{\mathbf{g}}_{i1}, \widehat{\mathbf{g}}_{i2}\}$ be the predicted offsets in the same cell produced by the network. Their computations are consistent with Eq. (7). If a positive cell contains only one circumcenter (*i.e.* $\tau = 1$), we consider it as having two identical circumcenters inside. If it contains $\tau \geq 2$ circumcenters, we match the two predictions to the ground-truth circumcenters that contribute to the minimum loss. The localization error is defined as an average of the smooth L1 loss between all pairs of matched $\mathbf{g}, \widehat{\mathbf{g}}$, *i.e.*

$$\mathcal{L}_2 = \frac{1}{N_p}\sum_{i=1}^{N_p} \min_{(a,b)\in P(\tau,2)} \left(\text{smooth}_{L1}(\mathbf{g}_{ia},\widehat{\mathbf{g}}_{i1}) + \text{smooth}_{L1}(\mathbf{g}_{ib},\widehat{\mathbf{g}}_{i2})\right). \tag{9}$$

$P(\tau, 2)$ represents the permutation of all 2 selections from $\tau$ elements, *e.g.*, $P(2,2) = \{(1,2),(2,1)\}$. For the special case of $\tau = 1$, we define $P(1,2) = \{(1,1)\}$. The $\text{smooth}_{L1}(\mathbf{g}_i, \widehat{\mathbf{g}}_i)$ is calculated by summing the scalar losses from each dimension of $\mathbf{g}_i$.

Eventually, the multi-task loss function of our neural network is formulated as $\mathcal{L} = \mathcal{L}_1 + \lambda\mathcal{L}_2$, where $\lambda$ is a hyperparameter for balancing the different loss terms.

### 3.5 INFERENCE

We train the network on local patches, but performing the inference on the complete point cloud. After predicting the adjacent triangles of each point, we take their union to form the primitive trian-

gulation (mesh). Since the primitive mesh is produced regardless of topological constraint, we apply post-processing to make it edge-manifold and fill the small holes. For convenience, we provide C implementations with Python interface for the post-processing.

# 4 EXPERIMENT

**ABC dataset.** The ABC dataset (Koch et al. 2019) is a collection of one million CAD models for deep geometry research. It provides clean synthetic meshes with high-quality triangulations. We use the first five chunks in this dataset to create the training and test sets. Each raw mesh is normalized into a unit sphere and decimated with a voxel grid of $0.01$. This results in a total of $9,026$ meshes, and we apply a train/test split of $25\%/75\%$ to validate effectiveness of the proposed model. The model is trained on the ABC training set. We assess it on the ABC test set as well as other unseen datasets. The implementation details are discussed in the supplementary.

**Evaluation criteria.** We evaluate the overall surface quality of each reconstructed mesh using Chamfer distances (CD1, CD2), F-Score (F1), normal consistancy (NC), and normal reconstruction error (NR) in degrees. We also evaluate their preservation of sharp details on the surface using Edge Chamfer Distance (ECD1) and Edge F-score (EF1), similar to (Chen et al., 2022). See the supplementary for computational details about those surface quality metrics. To compare the triangulation efficiency of learning-based methods, we report their inference time on the same machine. The number of points in the point cloud is provided for reference.

## 4.1 PERFORMANCE

**ABC test.** Table 1 compares the performance of the proposed method, abbreviated as 'CircNet', with those of the traditional triangulation methods, *i.e.,* $\alpha$-shapes (Edelsbrunner & Mücke 1994) and ball-pivot (Bernardini et al. 1999), the implicit surface method PSR (Kazhdan & Hoppe 2013) and the learning-based triangulation methods (*i.e.* PointTriNet (Sharp & Ovsjanikov 2020), IER (Liu et al. 2020), DSE (Rakotosaona et al. 2021b). We use the pre-trained weights of other learning-based methods to report their results. It is observed that training those methods from scratch on our ABC training set leads to slightly worse performance. We report the performance of $\alpha$-shapes using $\alpha = 3\%$ and $\alpha = 5\%$. We note that ball-pivot requires normals to be estimated first. The performance of PSR is reported by using normals $\mathbf{n}$ estimated from the point cloud, and normals $\mathbf{n}_{gt}$ computed from the ground-truth mesh. In the efficiency report, we also provide the network inference time of CircNet in brackets [.]. It can be seen that the proposed CircNet is much faster than the other learning methods. Besides, its reconstructed meshes are in high quality.

**Generalization.** We validate generalization of the proposed CircNet using unseen datasets. Those include FAUST (Bogo et al. 2014), a dataset of watertight meshes of human bodies; MGN (Bhatnagar et al. 2019), a dataset of open meshes of clothes; and several rooms of Matterport3D (Chang et al. 2017), a dataset of large-scale scenes reconstructed from RGB-D sequences. In all circumstances, the learning-based methods are evaluated without any fine-tuning. We report the results on FAUST and MGN in Table 2 and Table 3, respectively. It can be noticed that CircNet outperforms the other approaches most of time, especially in the reconstruction of the open surfaces from MGN. We observe that IER prefers uniform point clouds as inputs. Due to the non-uniformity of point clouds in FAUST and MGN, its performance drops significantly. As a reference, we provide its results on the uniformly resampled point clouds generated by Poisson-disk sampling (Bridson 2007).

Table 1: Surface quality of different triangulation methods on the ABC test set. For each metric, we report the average results across all meshes. The total triangulation time of each learning-based method is reported on the largest point cloud whose size is provided for reference. We also report the network inference time of CircNet in brackets [.].

| Method | Surface Quality | | | | | | | Efficiency | |
|---|---|---|---|---|---|---|---|---|---|
| | overall | | | | | sharp | | max | total |
| | CD1($\times 10^2$)↓ | CD2($\times 10^5$)↓ | F1↑ | NC↑ | NR↓ | ECD1($\times 10^2$)↓ | EF1↑ | #points | runtime/seconds |
| $\alpha$-shapes-3% | 0.448 | 2.670 | 0.836 | 0.943 | 7.203 | 2.628 | 0.616 | | 6.555 |
| $\alpha$-shapes-5% | 0.601 | 6.972 | 0.802 | 0.929 | 8.625 | 3.767 | 0.572 | | 6.544 |
| ball-pivot (+**n**) | 0.297 | 0.684 | 0.939 | 0.981 | 2.244 | 0.782 | 0.873 | | **0.347** |
| PSR (+**n**) | 0.403 | 6.700 | 0.894 | 0.971 | 6.493 | 32.402 | 0.095 | | 12.571 |
| PSR (+$\mathbf{n}_{gt}$) | 0.400 | 6.081 | 0.901 | 0.972 | 6.020 | 26.160 | 0.108 | 19669 | 12.529 |
| DSE | 0.285 | 0.548 | 0.949 | 0.985 | 1.793 | **0.538** | **0.929** | | 59.605 |
| IER | 0.289 | 0.580 | 0.945 | 0.983 | 1.949 | 0.890 | 0.914 | | 37.683 |
| PointTriNet | 0.288 | 0.790 | 0.948 | 0.984 | 1.931 | 0.688 | 0.926 | | 38.063 |
| CircNet (Prop.) | **0.284** | **0.544** | **0.950** | **0.985** | **1.758** | 0.708 | 0.924 | | 3.316 [0.996] |

Table 2: Method comparison on the watertight meshes of FAUST dataset. The 100 point clouds in this dataset each have 6890 points. We report the average runtime of each method per sample.

| Method | Surface Quality | | | | | | | Efficiency | |
| | overall | | | | | sharp | | #points | total |
| | CD1($\times 10^2$)↓ | CD2($\times 10^5$)↓ | F1↑ | NC↑ | NR↓ | ECD1($\times 10^2$)↓ | EF1↑ | | runtime/seconds |
|---|---|---|---|---|---|---|---|---|---|
| $\alpha$-shapes-3% | 0.551 | 3.689 | 0.757 | 0.894 | 18.197 | 7.222 | 0.087 | | 0.684 |
| $\alpha$-shapes-5% | 1.225 | 19.779 | 0.531 | 0.807 | 27.286 | 7.879 | 0.056 | | 0.681 |
| ball-pivot (+n) | 0.323 | 1.002 | 0.923 | 0.970 | 6.037 | 2.887 | 0.184 | | **0.138** |
| PSR (+n) | 1.119 | 39.229 | 0.564 | 0.863 | 21.721 | 4.161 | 0.438 | | 10.674 |
| PSR (+$n_{gt}$) | 0.427 | 4.108 | 0.915 | 0.969 | 10.269 | 1.069 | 0.810 | 6890 | 10.643 |
| DSE | **0.218** | **0.307** | **0.995** | **0.984** | **3.910** | **0.883** | 0.801 | | 23.792 |
| IER | 4.649 | 339.565 | 0.160 | 0.786 | 31.195 | 2.081 | 0.376 | | 13.949 |
| IER (Poisson) | 0.257 | 0.406 | 0.989 | 0.973 | 8.692 | 2.170 | 0.456 | | 10.628 |
| PointTriNet | 0.219 | 0.308 | 0.995 | 0.983 | 4.393 | 1.233 | 0.807 | | 13.344 |
| CircNet (Prop.) | 0.221 | 0.316 | 0.993 | 0.980 | 4.557 | 0.939 | **0.820** | | 3.471 [0.382] |

Table 3: Method comparison on the open surfaces of MGN dataset.

| Method | Surface Quality | | | | | | | Efficiency | |
| | overall | | | | | sharp | | max | total |
| | CD1($\times 10^2$)↓ | CD2($\times 10^5$)↓ | F1↑ | NC↑ | NR↓ | ECD1($\times 10^2$)↓ | EF1↑ | #points | runtime/seconds |
|---|---|---|---|---|---|---|---|---|---|
| $\alpha$-shapes-3% | 0.517 | 2.687 | 0.757 | 0.927 | 14.797 | 11.976 | 0.046 | | 1.399 |
| $\alpha$-shapes-5% | 0.899 | 10.322 | 0.578 | 0.883 | 20.233 | 14.227 | 0.023 | | 1.390 |
| ball-pivot (+n) | 0.462 | 4.917 | 0.844 | 0.974 | 5.803 | 11.847 | 0.083 | | **0.279** |
| PSR (+n) | 1.319 | 18.542 | 0.356 | 0.892 | 18.329 | 12.286 | 0.071 | | 10.389 |
| PSR (+$n_{gt}$) | 1.077 | 10.481 | 0.402 | 0.948 | 12.224 | 7.912 | 0.137 | 10116 | 10.623 |
| DSE | 0.270 | 0.530 | 0.968 | **0.983** | **3.970** | 4.508 | 0.440 | | 32.433 |
| IER | 0.827 | 23.464 | 0.796 | 0.972 | 6.220 | 4.932 | 0.447 | | 12.143 |
| IER (Poisson) | 0.310 | 0.635 | 0.948 | 0.980 | 7.073 | 6.777 | 0.317 | | 13.279 |
| PointTriNet | 0.272 | 0.562 | 0.967 | 0.981 | 4.398 | 5.936 | 0.399 | | 19.906 |
| CircNet (Prop.) | **0.269** | **0.512** | **0.968** | 0.981 | 4.230 | **3.231** | **0.486** | | 4.536 [0.535] |

Table 4: Our results on Matterport3D, as well as the uniform, non-uniform and noisy data.

| Data | | Surface Quality | | | | | | | Efficiency | |
| | | overall | | | | | sharp | | #points | total |
| | | CD1($\times 10^2$)↓ | CD2($\times 10^5$)↓ | F1↑ | NC↑ | NR↓ | ECD1($\times 10^2$)↓ | EF1↑ | | runtime/seconds |
|---|---|---|---|---|---|---|---|---|---|---|
| Matterport3D | | 0.151 | 0.144 | 0.999 | 0.933 | 10.870 | 0.249 | 0.974 | $5 \times 10^5$ | ∼9.7 minutes |
| Robustness (ABC) | poisson | 0.278 | 0.533 | 0.949 | 0.976 | 4.033 | 1.281 | 0.660 | $10^4$ | 3.190 [0.562] |
| | $\sigma = 0.1$ | 0.328 | 0.720 | 0.908 | 0.965 | 10.053 | 1.500 | 0.630 | $10^4$ | 3.368 [0.499] |
| | $\sigma = 0.2$ | 0.419 | 1.245 | 0.793 | 0.931 | 16.735 | 4.396 | 0.467 | $10^4$ | 5.921 [0.549] |
| | $\sigma = 0.3$ | 0.523 | 2.076 | 0.689 | 0.880 | 23.287 | 6.790 | 0.294 | $10^4$ | 9.776 [0.627] |
| | non-uniform | 0.395 | 1.852 | 0.858 | 0.942 | 8.693 | 3.490 | 0.457 | 5085 (mean) | 1.958 [0.525] |

For Matterport3D, we report the quantitative results of CircNet over 25 scenes in Table 4. The input point clouds are generated by uniformly sampling $5 \times 10^5$ points from the raw meshes. We compare the reconstructed meshes of CircNet with others on a large building in Fig. 5.

**Robustness.** To test the robustness of CircNet, we generate point clouds that are uniform, noisy and non-uniform using 100 meshes of the ABC test set. Specifically, we apply the Poisson disk sampling to generate uniform point clouds with $10^4$ points each. We also add different levels of Gaussian noise (*i.e.*, $\sigma = 0.1, 0.2, 0.3$) to the data to obtain noisy point clouds. To get non-uniform point clouds, we vary the densities of the uniform data along an axis, similar to DSE (Rakotosaona et al. 2021b). The quantitative results of CircNet for each category are provided in Table 4. We report the quantitative comparisons in the supplementary. Figure 5 shows the reconstructed meshes of CircNet for point clouds of varying densities or with noise using an interesting shape from the Thingi10K dataset (Zhou & Jacobson 2016). The point cloud data are generated in similar ways.

**Limitations.** We analyze detection accuracy of different triangles according to their maximum interior angles. Figure 6 plots such results for the ABC and FAUST datasets. Without loss of generality, we randomly sample 500 meshes in ABC to plot the results. We compute the detection accuracy as the ratio between the number of detected triangles and the ground-truth number. When the maximum interior angle becomes large, the radii of the triangle circumcircles can be in an *intractable* ($\rightarrow +\infty$) range. This makes the circumcenter detection difficult using fixed anchor priors. As expected, the detection accuracy decreases with the increase of the maximum angle. So far, handling edge-manifoldness takes non-trivial amount of time in all learning-based triangulation methods. This is because the classification or detection networks are unaware of any surface topology.

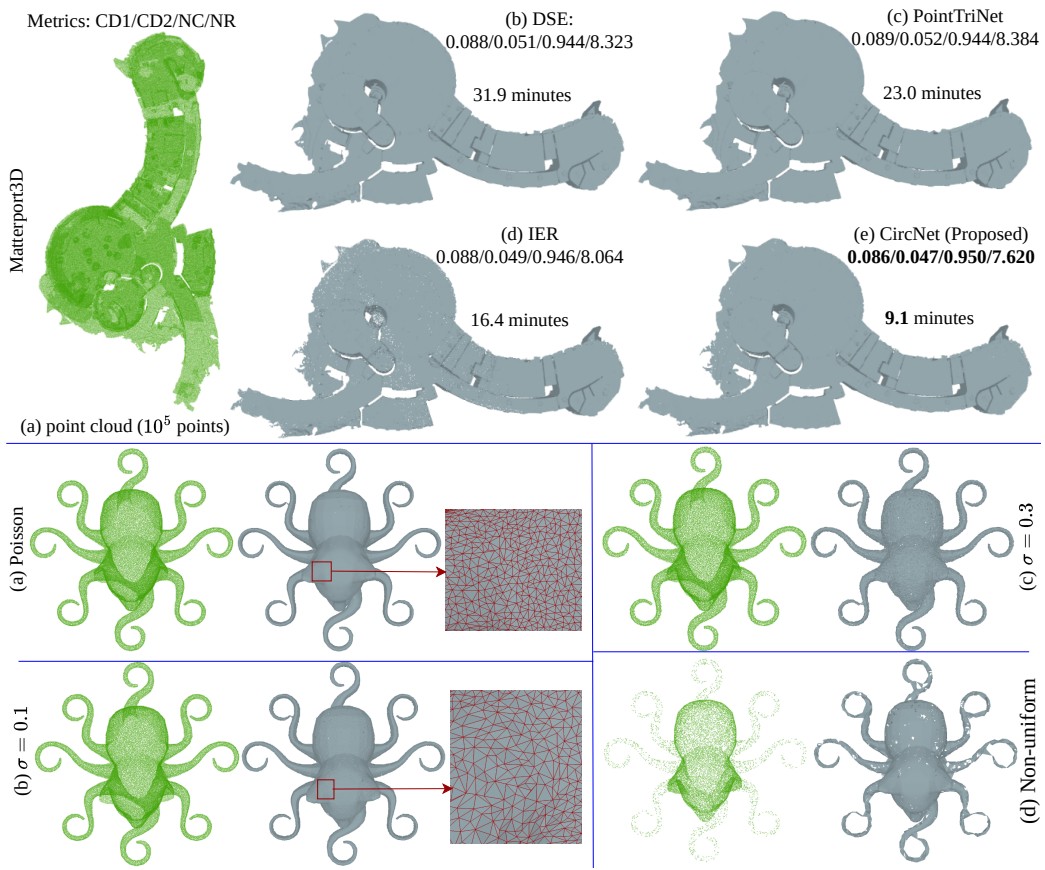

Figure 5: Visualization of the reconstructed meshes. Quantitative results of CD1/CD2/NC/NR and the reconstruction time are reported for different methods over a scene of Matterport3D. The proposed CircNet takes the shortest time but reconstructs the scene in best quality. The bottom two rows demonstrate the robustness of CircNet for the fact that it still reconstructs the underlying shape well when the point cloud becomes highly noisy or the non-uniformity becomes severe.

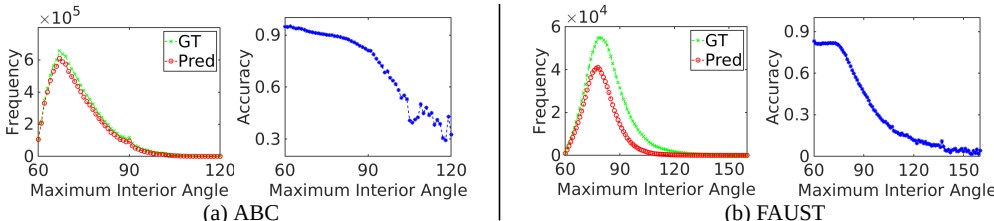

Figure 6: Detection frequency and accuracy with respect to the maximum interior angles (in degree) of ground-truth triangles. We provide results on two datasets, ABC and FAUST. For each dataset, the left plot shows the frequencies of both ground-truth and correctly predicted triangles, while the right plot shows the prediction accuracy. It can be noticed that triangles in ABC are distributed more closely to equilateral triangles, compared to those in FAUST.

## 5 CONCLUSION

By exploiting the duality between a triangle and its circumcenter, we have introduced a neural detection architecture for point cloud triangulation. The proposed network employs a single-shot pipeline, and takes local geometrics as inputs to detect the circumcenters of adjacent triangles of each point. We predefine multiple anchor points to guide the detection process. Based on the detected circumcenters, we reconstruct the 3D point cloud into a primitive triangle mesh. It can be simply post-processed into a surface mesh. Compared to the previous learning-based triangulation methods, the proposed method has lower complexity, single-shot architecture, and does not depend on any traditional method to triangulate the points. We demonstrate that the method, though trained on CAD models, is able to reconstruct unseen point clouds including the large-scale scene data satisfactorily. It is also robust to noisy and non-uniform data.

## 6 ACKNOWLEDGEMENT

This research is funded in part by the ARC Discovery Grant DP220100800 to HL. We thank the anonymous reviewers for their comments on improving the quality of this work.

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
