# OpenReview forum: "CircNet: Meshing 3D Point Clouds with Circumcenter Detection"
_ICLR.cc/2023/Conference — ICLR 2023 poster_

### Official Review · Reviewer_Sk4W · 2022-10-20

**Confidence:** 4
**Correctness:** 4
**Technical Novelty And Significance:** 3
**Empirical Novelty And Significance:** 3
**Recommendation:** 8

**Clarity, Quality, Novelty And Reproducibility:**

Overall, this paper is clearly written and easy to follow. The idea of reformulating the point cloud trianguation problem into a circumcenter detection problem in a deep learning setting is original. The quantitative and qualitative results reported in the paper are both convincing and promising. There are also sufficient details included in the paper to reproduce the results.

**Strength And Weaknesses:**

Strength
+ The formulation of the problem as circumcenter detection sounds novel in deep learning based point cloud triangulation.
+ The formulation of the problem as circumcenter detection is technically sound.
+ The proposed method does not require exhaustive enumeration of traingle combinations or local surface parameterization, which are computationally expensive procedures.
+ The proposed method does not require any surface normal information, which are typically not available for a point cloud.
+ The proposed method is a single-stage pipeline and has high efficiency compared to exsiting two-stage pipelines.
+ The proposed method has achieved SOTA results in term of quality in the ABC dataset and MGN dataset, and highly competitive results in the FAUST dataset.
+ The proposed method has achieved SOTA results in term of efficieny among deep learning based methods.
+ The proposed method is robust to noisy and non-unifrom data.
+ The proposed method works for both water-tight and open meshes.

Weaknesses
- The anchor based detection approach is not novel but borrowed from existing object detection networks. Novelty is limited in this technical aspect.
- There is no evaluation on the effect of K (in KNN patch composition) and s (in detecting multiple circumcenters in a single anchor cell).

**Summary Of The Paper:**

This paper tackles the problem of point cloud triangulation. The authors introduce a deep neural network to detect circumcenters from point patches, and then they extract triangles dual to the detected circumcenters to form a primiteive mesh. The proposed method can reconstruct meshes within a couple of seconds, and poses no restriction on the surface topology. Evaluation in terms of quality, efficiency, and robustness has been carried out on 4 prominent datasets.

**Summary Of The Review:**

The idea of detecting circumcenters from point cloud patches and extracting traingles dual to the circumcenters is technically sound and novel. Sufficient evalations have been carried out to demonstrate the effectiveness, efficiency, and the robsutness of the proposed method. Experimental results show that the proposed single-stage pipeline outperforms existing two-stage pipelines in terms of reconstruction quality and efficiency. I would support accepting this paper.

---

> ### Author Response · Authors · 2022-11-18
> **Author Responses to Reviewer #4(Sk4W)**
>
> **Reviewer 4.1:** We are grateful to the reviewer for the high-quality review. We thank the reviewer for clearly summarising the strengths of our work and commenting our dual reformulation of point cloud triangulation as original.
>
> **Reviewer 4.2:** There are two options to achieve a detection task, anchor-guided or anchor-free. We argue that the detection of a *point* can be much more challenging than the detection of a rectangular box with area. We choose the anchor-guided solution as it facilitates the detection by constraining each circumcenter to be inside a small subspace. The proposed CircNet is trained under the supervision of L1 loss and cross-entropy, which are essential loss functions for detection tasks. Our limitation now comes in the absence of explicit topology-based losses to supervise the model training, as is discussed in the paper. We are now exploring those possibilities, which can result in an advanced and more efficient model for point cloud triangulation. Nevertheless, the performance of CircNet is already much better than we expected in the very beginning of this work. It demonstrates that detecting circumcenters to triangulate point clouds is indeed possible and very promising.
>
> **Reviewer 4.3:** We thank the reviewer for recommending  evaluations on the effect of neighborhood size ($K$) and the number of predictions per anchor cell ($s$). We have compared the model performance of $K=50$ to that of $K=25,100,200$. It is noticed that $K=100$ and $K=200$ perform similarly to $K=50$, while $K=25$ results in performance drop. The reason is that $K=25$ is too small to cover the 1-ring neighborhood. Given similar performance, smaller $K$ is preferred for better efficiency. As for $s$, we do not recommend settings of $s>2$ because dominantly, each anchor cell is observed to be populated by at most two ground-truth circumcenters. We hence compare the model performance of $s=2$ to that of $s=1$ only. It shows that the overall surface quality of $s=2$ is slightly better. All of these additional experiments have been summarised in Table G of the supplementary of our revised paper.

---

### Official Review · Reviewer_7M62 · 2022-10-22

**Confidence:** 4
**Correctness:** 2
**Technical Novelty And Significance:** 2
**Empirical Novelty And Significance:** 1
**Recommendation:** 3

**Clarity, Quality, Novelty And Reproducibility:**

The paper is mostly clearly written and easy to follow. Most of the detail necessary to reproduce the paper seems to appear in the main text or the supplementary material. The attached code includes the network implementation and validation code - however it does not include the code for training the network, or the code for post-processing the network output (local per-point triangulations) into a global triangulation.

**Strength And Weaknesses:**

Strengths
* The paper is clearly written and easy to follow.
* The paper proposes a novel method to predict local per-point triangulation of a point cloud with a graph neural network.

Weaknesses
* The proposed deep learning method only predicts local per-point triangulation (1-ring neighborhood). Even this steps does not provide any guarantees of the validity or quality of the local triangulation - e.g. can the triangles overlaps? Some experiments and metrics need to be added to illustrate and validate this step.
* The main weakness of the proposed method is the fact that it does not discuss in any detail how these per-point triangulations are post-processed into a valid (e.g. non-overlapping triangles) and useable (e.g. for rendering) triangulation of the whole point cloud. Since nothing in the network design constraints neighboring points to have consistent triangulations after the inference, the postprocessing step seems to be essential to fix the inconsistencies. The paper also does not provide any metrics and almost no visualizations to illustrate the validity and useability of the reconstructed meshes. In addition, it is mentioned in the paper that a C implementation of the postprocessing is provided, however it does not appear in the attached code submission.

**Summary Of The Paper:**

The paper proposes a method for point cloud triangulation by creating local per-point triangulations and then post-processing these local triangulations into a global point cloud triangulation. The first step is performed using a graph neural network and it is the main contribution of the paper. Specifically, the paper proposes to predict the presence and the location of the circumcenters of the triangles around each point using a graph neural network. The paper further proposes to discretize the space around each point into predefined anchor cell and predict the circumcenters inside these anchor cells.

**Summary Of The Review:**

* The attached code does not include the code for training the model, or for post-processing the network output (local per-point triangulations) into a global triangulation.

* Not all parameter choices are explained - e.g. how is M chosen (Sec. 3.1).

* In section 3.1 and Figure 4: what constraints the network to produce valid local triangulations? E.g. that the triangles in this 1-ring neighborhood of a point are non-overlapping. Provide some quantitative experiments.

* I suggest to move Eq. (7) and the preceding sentence starting with "The proposed network detects each circumcenter" to section 3.2, to explain how anchors are used to predict circumcenters. Also, use “hat” above "g"s in Eq. (7) to specify that these are offsets from predicted circumcenters, not from the ground truth ones.

* At the end of Section 3.5, the paper states "Since the primitive mesh is produced regardless of topological constraint, we apply post-processing to make it edge-manifold and fill the small holes. For convenience, we provide C implementations with Python interface for the post-processing." This post-processing seems like a very important part of the proposed method. Since nothing constraints neighboring points to have consistent triangulations after the inference, the postprocessing step is essential to fix the inconsistencies and to produce consistent and useable point cloud triangulation. For such an important step, the two line explanation which does not contain any detail about the method is not sufficient. Also, as stated above, the code linked to the submission does not include the postprocessing code.

* Evaluation criteria: Elaborate on why CD1, CD2, F1 are the right metrics here. E.g. what exactly does Chamfer distance measure here and how it is related to the quality of the triangulation. Add an evaluation metric measuring the quality of the constructed meshes in terms of how well they approximate 2D surfaces.

* Figure 5:
    * (a) Add visualization of a small parts of the mesh showing the quality of the triangulation. Render the resulting mesh to show the triangulation quality visually.
    * (d) The triangulation seems to contain holes for non-uniformly sampled point cloud, contradicting the statement in the caption that the method is robust to variations in the point cloud density.
    * The caption is very lengthy and can be improved by moving come explanations to the main text.

---

> ### Author Response · Authors · 2022-11-18
> **Author Responses to Reviewer #3(7M62): Part A**
>
> **Reviewer 3.1:** We thank the reviewer for the careful review. Our main contribution is reformulating point cloud triangulation as a dual problem of circumcenter detection. It enables the proposed method to avoid triangle enumerations as well as local surface parameterization, which are computationally expensive. Due to the reformulation, we successfully reduce the computational complexity from $\mathcal{O}(K^2N)$ to $\mathcal{O}(tN)$. The proposed triangulation is based on a neural network of only 5 layers, which is much shallower than those used by DSE ($2\times13$ layers), IER (19 layers) and PointTriNet (21 layers). Reviewer \#4 (Sk4W) clearly identifies our contribution and commented it as *original*.
>
> **Reviewer 3.2:** As discussed in Sec. 3.5, our post-processing is composed of edge-manifold control and filling small holes. Both are *standard* operations for meshes in geometry processing. We therefore do not elaborate on their background details. This is also the choice of previous methods, IER and PointTriNet. For explicity, we have shared the post-processing code in the anonymous link.
>
> **Reviewer 3.3:** For review purpose, we only released the evaluation code. The anonymous link now contains complete codes for our network training/evaluation, post-processing, and computations of all metrics to assess the surface quality.
>
> **Reviewer 3.4:** The comment ``this paper does not provide any metrics to illustrate validity of the reconstructed meshes" is *incorrect*. In the comparison of all tables, we have reported the surface quality in terms of Chamfer distances, F-score and Normal Consistency, which are altogether standard metrics for evaluating how well a reconstructed mesh approximates the original manifold surface. We have discussed those evaluation criteria in the supplementary. On the reviewer's request and for explicity, we provide the definitions of Chamfer distance and F-score in the supplementary of our revised paper. The code of all metric computations is shared in our anonymous link. We thank the reviewer for the query on CD1, CD2 and F1.
>
> **Reviewer 3.5:** The comment ``this paper also does not provide visualizations for the reconstructed meshes'' is not true. In fact, to show the reconstruction quality of the proposed approach, we provide visualizations not only for the complex watertight shapes but also for the large-scale open surfaces. We also show the triangulation details on small parts of the clean and noisy shapes. Please see Fig. 5 of the main paper and Fig. B in the supplementary for the results.
>
> **Reviewer 3.6:** As agreed by the other reviewers, all of our claims and statements are well-supported and correct. We note that in Fig. 5, small holes in the reconstructed shape *is not* a contradiction of our robustness to non-uniform point clouds. On the contrary, it demonstrates the robustness of our method for the fact that, the method still reconstructs the complex underlying shape completely, and none of its 8 sparsely sampled tails are chopped out. This is challenging for many other approaches, *e.g.,* alpha-shape, ball-pivot, PSR, IER, DSE. Please see Fig. C of our revised supplementary for the visualized comparisons. We thank the reviewer for the comment on the caption of Fig. 5, which we have revised as advised.

---

> > ### Author Response · Authors · 2022-11-18
> > **Author Responses to Reviewer #3(7M62): Part B**
> >
> > **Reviewer 3.7:** The topology of our local triangulation is implicitly supervised by the ground-truth $K$NN patches, which are edge-manifold and without holes. To show the quality of our local predictions, we report additional metrics in terms of the average accuracy (mAcc) and average intersection-over-union (mIoU) of the $K$NN patches. Please see the supplementary of our revision (Fig. A) for the results. We note that in those computations, per triangle is a prediction element.
> >
> > Three kinds of overlaps can happen between the local triangulations. (1) The first one is duplicate triangle predictions. However, since we compute the primitive mesh by taking the union of all local triangles, as shown in Fig. 2, this kind of overlap does not exist in the primitive mesh. (2) The second type is triangles sharing a single edge. In an edge-manifold mesh, each edge is shared by at most two triangle faces. Please see the $1st$ paragraph of Sec. 3 for detailed explanations. We address this kind of overlap with edge-manifold post-processing. (3) The third type is triangles sharing a single vertex, which is referred as vertex-manifoldness in geometry processing.
> > It is beyond the concern of existing learning-based methods for point cloud triangulation.
> >
> > The ultimate goal of this paper is to propose a *complete* pipeline for triangulating 3D point clouds into edge-manifold meshes. The validity of our local triangulaions is demonstrated by our SOTA or comparable results on the ABC, MGN and FAUST datasets. We note that the proposed method not only generalizes well but also is robust. We thank the reviewer for the query on local triangulations.
> >
> > **Reviewer 3.8:** We only use $M$ in Sec 3.1 to denote the number of triangles in a reconstructed mesh. It is neither a trainable parameter nor hyper-parameter of our method. We use different subscripts for the ground-truth and the predicted circumcenter (*i.e.,*, ${\mathbf X}_i({\mathbf p}), \widehat{{\mathbf X}}_m({\mathbf p})$) as the number of triangles in the ground-truth and the reconstructed meshes $\mathcal{T},~\widehat{\mathcal{T}}$ can differ. All of the hyper-parameters required for our network training and evaluation have been reported in the original supplementary.
> >
> > **Reviewer 3.9:** The notations $g_i^\rho$, $g_i^\theta$, $g_i^\phi$ in Eq. (7) define parameterized offsets of the ground-truth circumcenter ${\mathbf X}_i({\mathbf p})$. Therefore, our using $g$ instead of $\hat{g}$ is actually *correct*. The predicted offsets of the neural network are represented as $(\hat{g}_m^\rho,\hat{g}_m^\theta,\hat{g}_m^\phi)$. We have slightly revised the texts around Eq. (7) for further clarity. In addition, we think it is better to keep Eq. (7) in Sec. 3.4 as it facilitates the discussion of matching strategy. We thank the reviewer for the suggestion and the careful review.

---

### Official Review · Reviewer_78GV · 2022-11-04

**Confidence:** 2
**Correctness:** 4
**Technical Novelty And Significance:** 3
**Empirical Novelty And Significance:** 2
**Recommendation:** 6

**Clarity, Quality, Novelty And Reproducibility:**

The paper is easy to follow, and the writing is OK, and the method can be reproduced.



**Strength And Weaknesses:**

Strength:
1) The paper is well-written and easy to follow.
2) The method is technically sound.
3) Although the method does not beat the state-of-the-art in runtime neither surface quality, authors show a significant speedup compared with learning-based baselines for good surface quality.

Weaknesses:
1) The method is still slow compared to the traditional baselines.

Issues/questions:
1) It would be good to define an "optimal triangle";
2) The authors mention that the final mesh is obtained by filling small holes in the primitive mesh. It is unclear to me why is this needed, and how are they being filled.
3) The authors use Anchor priors using spherical coordinates. The motivation for the use of spherical coordinates is not clear to me.

**Summary Of The Paper:**

The authors propose a method for creating a mesh from point clouds taking advantage of learning-based techniques.

The main contribution to the paper is the use of the duality relationship between the mesh triangle and the respective circumcenter, in which each triangle vertex is equally distant from the circumcenter. This is done using a deep neural network.


**Summary Of The Review:**

The paper has some interesting contributions and, although the proposed meshing is still slower than the traditional baselines, results show that it is significantly faster than previous learning-based methods.

---

> ### Author Response · Authors · 2022-11-18
> **Author responses to Reviewer #2(78GV)**
>
> **Reviewer 2.1:** We thank the reviewer for the review, and noticing our good reconstruction quality with significant speedup compared to the learning-based baselines.
>
> Regarding the comment ``the method is slow compared to the traditional baselines'', we argue that the traditional baselines, alpha-shape and ball-pivot, are unable to reconstruct the surface with acceptable quality, which is exactly the motivation for learning-based point cloud triangulation. Given the reconstruction quality of those methods, it is no longer fair to compare our runtime to theirs. Meanwhile, we note that the fast runtime of traditional methods also benefits from highly optimized implementations in the modern Library. Besides, ball-pivot requires point normals as input.
>
>
> **Reviewer 2.2:** It is difficult to define an optimal triangle. Although equilateral triangles are preferred in the triangulation, their existence is only possible for uniformly sampled point cloud. Yet, obtuse triangles with small angles (e.g. $10^\circ$) are generally considered as bad triangles in a triangulation.
>
>
> **Reviewer 2.3:** The small holes are because of missing triangles in the reconstructed mesh. They are different from the topological *genus* in a shape. In geometry processing, the operation of filling holes is to search for empty loops that comprise only boundary edges, and add the missing triangles back. We fill the small loops composed of three or four boundary edges. The post-processing of filling holes is actually fast. Our bottleneck now lies in the edge-manifold process.
>
> For review purpose, we only released the evaluation code. The anonymous link now contains complete codes for our network training/evaluation, post-processing, and computations of all metrics to assess the surface quality.
>
>
> **Reviewer 2.4:** There are three different methods to divide the local neighborhood space of a point, *i.e.* volumetric partitions, spherical partitions, and data clustering. We use spherical coordinates as its azimuth ($\theta$) and inclination ($\phi$) directions are bounded, *i.e.* $\theta\in(-\pi,\pi], \phi\in[-\frac{\pi}{2},\frac{\pi}{2}]$. We do not use the other two alternatives for they lead to higher computational complexity. Please see Lines 3-4 in the $1st$ paragraph of Section 3.2 and the texts below Eq. (5) of Sec. 3.2 for further explanations.

---

### Official Review · Reviewer_VP72 · 2022-11-06

**Confidence:** 3
**Clarity, Quality, Novelty And Reproducibility:** 1. Quality of results can be improved…
**Correctness:** 4
**Technical Novelty And Significance:** 3
**Empirical Novelty And Significance:** 3
**Recommendation:** 6

**Details Of Ethics Concerns:**

No ethical concerns

**Strength And Weaknesses:**

Strengths:-
1. A novel method to generate  triangular meshes for the the input 3D point cloud. First learning based technique to generate point cloud by estimating the cirumcenters in point local neighborhood space.
2. Method seems to generalize on real scenes with noisy data.
3. Method is much faster than other state of the art learning based techniques.
4. Theoretical foundations of the paper seem to be correct.

Weakness:-
1. The paper lacks to provide enough visual evidences of real benefits of the paper over standard point cloud triangulation techniques. In my opinion, the authors should make it very clear how is accuracy improving in comparison to other techniques. While, some metrics like chamfer dist. (and edge distance) speak for the accuracy, I believe visual aspect of highlighting the regions where accuracy or completeness is improved increases the actual impact and brings more clarity for the reader.  Authors can highlight regions where standard techniques struggle, for example, thin structures, bad triangulation (non-manifold errors).
2. Although, it is fair to say implicit surface representation works are fundamentally different than mesh triangulation but I guess comparing results with such methods might highlight the strengths/weakness of both classes of methods. This enables complete and higher level study to understand true value of choosing learning based mesh triangulation over something like Poisson surface reconstruction.


**Summary Of The Paper:**

The paper introduces a novel method to generate triangular meshes for the the input 3D point cloud. The authors introduce multiple anchors  to divide the neighborhood space of each input 3D point. A deep neural network is then used to predict the presence and the locations of cirumcenters in the point local neighborhood space. Then the dual of the circumcenter is used to extract triangular mesh.
Contributions:-
1. This paper shows that their method is more efficient and accurate than the state of the art learning based techniques on point cloud triangulation.
2. The proposed method generalizes on real scenes with noisy data and doesn't hold restrictions on mesh topology as in contrast to some other learning based techniques.


**Summary Of The Review:**

The paper introduces a novel method to generate triangular meshes for the the input 3D point cloud. The authors use a deep neural network  to predict the presence and the locations of circumcenters in the point local neighborhood space. Then the dual of the circumcenter is used to extract triangular mesh. The paper provides some evidence that their method is more efficient and accurate than the state of the art mesh triangulation techniques. The impact of the paper can be improved by improving the quality of visualization of the results by highlighting, the regions of improvements. Discussing the results with other class of methods such as implicit surface reconstruction ones will bring out true benefits and impact of such method.

---

> ### Author Response · Authors · 2022-11-18
> **Author Responses to Reviewer #1(VP72)**
>
> **Reviewer 1.1:** We thank the reviewer for recognizing the originality of our theoretical foundations, and the novelty, generalizability (on real scenes with noisy data) as well as  efficiency of our method.  We are grateful to the reviewer for the efforts he/she has made to improve the impact of our work.
>
>
> **Reviewer 1.2:** We provide highly-compressed visualizations in the submission for space concern. In the supplementary of our revision, more visualizations and quality comparisons have been provided as advised. We thank the reviewer for the detailed suggestions.
>
>
> **Reviewer 1.3:** We did not report the results of Poisson surface reconstruction (PSR) in our original submission,  because PSR assumes surface normals as *known*, which is typically not true. However, on the reviewer's request and for completeness,  in our revised paper, we have included those results of PSR on the ABC, FAUST and MGN datasets. Specifically, we report its different performance while using the ground-truth normals and estimated normals. The estimated normals are computed based on the 10-nearest neighbors of a point. We thank the reviewer for this suggestion.

---

### Author Response · Authors · 2022-11-18
**Summary of Our Paper Revisions**

*We thank all the reviewers for their comments and efforts to make this work better.* All the revisions are highlighted in **red** texts. For tables, we highlight their captions if there are changes.

**Based on the comments of Reviewer #1(VP72)**, we have (1) included the performance of the classic Poisson surface reconstruction method (PSR) in Table 1, Table 2 and Table 3; (2) added more visualisations about quality comparison.

**Based on the comments of Reviewer #3(7M62)**, we have (1) discussed the Chamfer distances and F-score in details in the supplementary of our revised paper; (2) updated the texts around Eq. (7), and revised the caption of Fig. 5.

**Based on the comments of Reviewer #4(Sk4W)**, we have analysed the effects of different hyper-parameters ($K$, $s$) on our model performance.

---

### Decision · Program_Chairs · 2023-01-20

**Decision:**

Accept: poster

**Justification For Why Not Higher Score:**

A relatively small subfield of interested practitioners.

**Justification For Why Not Lower Score:**

Sufficient novelty, potentially useful within that subfield.

**Metareview: Summary, Strengths And Weaknesses:**

Problem: Given a cloud of 3D points, without associated surface normals, recover a triangle mesh.

Contribution: an interesting hybrid of ML model (GNN to predict "circumcenters" from local point networks) and conventional mesh manipulation.  Presents good results compared to other systems, and is a useful contribution to this subfield.  Reviewers agree that the idea is novel/original.

Weaknesses (post rebuttal):  It remains the case that the postprocessing is somewhat ad-hoc, and the relative contribution of the various components is hard to discriminate, either theoretically or empirically.



**Note From Pc:**

if the above contains the word "oral" or "spotlight" please see: "oral" presentation means -> notable-top-5% and "spotlight" means -> notable-top-25%. As stated in our emails, we are disassociating presentation type from AC recommendations

**Summary Of Ac-Reviewer Meeting:**

The reviewers' primary concerns before rebuttal were:

The postprocessing code was not included.  This was provided for rebuttal, and studied in the meeting.  It was agreed with the rebuttal text that it represents "standard" mesh-wrangling code, although it is strongly recommended that the authors perform some ablations, e.g. run the postprocessing code on the raw point cloud at various densities.

Too few visual results.  The rebuttal contains new and useful visual comparisons, which were considered in the meeting.  These results illustrate that the proposed method exceeds existing methods in several places, despite some lack of smoothness in others.  In particular, performance in the "no noise" regime appears poorer than other methods.  Of course, this is not an important regime in practice, but it would be instructive to see a discussion or exploration of this feature of the algorithm.